# Multidrug-Resistant (MDR) Urinary Tract Infections Associated with Gut Microbiota in CoV and Non-CoV Patients in a Urological Clinic during the Pandemic: A Single Center Experience

**DOI:** 10.3390/antibiotics12060973

**Published:** 2023-05-28

**Authors:** Viorel Dragos Radu, Radu Cristian Costache, Pavel Onofrei, Egidia Miftode, Iacov Linga, Radu Mihaita Ouatu, Lucian Boiculese, Razvan Lucian Bobeica, Ingrid Tanasa Vasilache, Irina Luanda Mititiuc

**Affiliations:** 1Department of Urology, Faculty of Medicine, University of Medicine and Pharmacy “Gr. T. Popa”, 700115 Iasi, Romania; viorel.radu@umfiasi.ro (V.D.R.); radu.costache@umfiasi.ro (R.C.C.); 2Department of Urology and Renal Transplantation, “C.I. Parhon” University Hospital, 700115 Iasi, Romania; iacov_linga@email.umfiasi.ro (I.L.); radu_oatu@email.umfiasi.ro (R.M.O.); bobeica.razvan-lucian@d.umfiasi.ro (R.L.B.); 3Department of Morpho-Functional Sciences II, Faculty of Medicine, University of Medicine and Pharmacy “Gr. T. Popa”, 700115 Iasi, Romania; 4Department of Infectious Diseases (Internal Medicine II), Faculty of Medicine, University of Medicine and Pharmacy “Gr. T. Popa”, 700115 Iasi, Romania; egidia.miftode@umafiasi.ro; 5Department of Preventive and Interdisciplinarity, Medical Informatics and Biostatistics, Faculty of Medicine, University of Medicine and Pharmacy “Gr. T. Popa”, 700115 Iasi, Romania; vasile.boiculese@umfiasi.ro; 6Department of Obstetrics and Gynaecology, Faculty of Medicine, University of Medicine and Pharmacy “Gr. T. Popa”, 700115 Iasi, Romania; ingrid-andrada.vasilache@umfiasi.ro; 7Department of Internal Medicine II, Faculty of Medicine, University of Medicine and Pharmacy “Gr. T. Popa”, 700115 Iasi, Romania; irina.mititiuc@umfiasi.ro

**Keywords:** multidrug resistant bacteria, urinary infections, COVID-19 infection

## Abstract

The aim of the study was to compare the profile of COVID-19 (CoV)-infected patients with non-COVID-19 (non-CoV) patients who presented with a multidrug-resistant urinary tract infection (MDR UTI) associated with gut microbiota, as well as analyze the risk factors for their occurrence, the types of bacteria involved, and their spectrum of sensitivity. Methods: We conducted a case–control study on patients admitted to the urology clinic of the “Parhon” Teaching Hospital in Iasi, Romania, between March 2020 and August 2022. The study group consisted of 22 CoV patients with MDR urinary infections associated with gut microbiota. For the control group, 66 non-CoV patients who developed MDR urinary infections associated with gut microbiota were selected. Electronic medical records were analyzed to determine demographics, characteristics, and risk factors. The types of urinary tract bacteria involved in the occurrence of MDR urinary infections and their sensitivity spectrum were also analyzed. Results: Patients in both groups studied were over 60 years of age, with no differences in gender, environment of origin, and rate of comorbidities. Patients in the CoV group had a higher percentage of urosepsis (54.5% versus 21.2%, *p* < 0.05) and more hospitalization days (9.27 versus 6.09, *p* < 0.05). Regarding risk factors, the two groups had similar percentages of previous urologic interventions (95.45% versus 96.97%, *p* > 0.05), antibiotic therapy (77.3% versus 87.9%, *p* > 0.05), and the presence of permanent urinary catheters (77.27% versus 84.85%, *p* > 0.05). *Escherichia coli* (31.8% versus 42.4%, *p* > 0.05), *Klebsiella* spp. (22.7% versus 34.8%, *p* > 0.05), and *Pseudomonas aeruginosa* (27.3% versus 9.1%, *p* > 0.05) were the most common urinary tract bacteria found in the etiology of MDR urinary infections in CoV and non-CoV patients. A high percentage of the involved MDR urinary tract bacteria were resistant to quinolones (71.4–76.2% versus 80.3–82%, *p* > 0.05) and cephalosporins (61.9–81% versus 63.9–83.6%, *p* > 0.05), both in CoV and non-CoV patients. Conclusions: Patients with urological interventions who remain on indwelling urinary catheters are at an increased risk of developing MDR urinary infections associated with gut microbiota resistant to quinolones and cephalosporins. Patients with MDR UTIs who have CoV-associated symptoms seem to have a higher rate of urosepsis and a longer hospitalization length.

## 1. Introduction

Urinary tract infections (UTIs) associated with gut microbiota represent the most common multidrug-resistant (MDR) infections encountered in urology departments. The increase in antibiotic resistance and the occurrence of urinary tract infections associated with MDR gut microbiota represent an important public health problem that increases patient morbidity and mortality [1]. According to some authors, the CoV pandemic led to a decrease in the incidence of MDR UTIs [2,3], which can be attributed to strict isolation and hygiene measures, whereas other authors found similar rates of occurrence of MDR UTIs [4,5,6]. There is limited information on the characteristics of MDR UTIs in CoV patients, as well as on how the COVID-19 pandemic altered the profile of MDR UTI patients. In addition, we have insufficient data about the impact of the pandemic on risk factors, the types of urinary tract bacteria and their sensitivity spectrum, and whether there are differences between CoV patients and non-CoV patients who have these types of infections. Therefore, we conducted a study to analyze all these factors to help urologists and infectious disease physicians actively detect MDR UTIs according to patient profiles and initiate early antibiotic treatments against these microorganisms.

## 2. Results

Between March 2020 and August 2022, 95 urological patients were hospitalized with a diagnosis of CoV. Among them, 27 had UTIs, of which 22 (23.16%) patients had UTIs associated with MDR microorganisms from the gut microbiota, which formed the study group. During the same period, 66 non-CoV patients with MDR UTIs were consecutively enrolled as a control group. In the CoV group, 20 patients were diagnosed with MDR UTIs via a urine culture taken during hospitalization in the first hours, and two patients were diagnosed with MDR UTIs via a urine culture on days 7 and 12, respectively, after an initial negative urine culture upon admission. In 20 patients, the diagnosis of CoV was made before or at the time of hospitalization. In two cases, the diagnosis was made on the 5th and the 12th day of hospitalization, respectively, despite a negative PCR test upon admission. In the control group, the MDR UTI diagnosis was made via a urine culture performed at the time of admission. In the control group, there was a patient with acute left orchiepididymitis and with an MDR UTI. He had not been hospitalized previously, had not taken antibiotics in the past, and had not undergone urological procedures. Otherwise, all patients in the CoV and non-CoV groups had a history of hospitalization or urological maneuvers. Table 1 shows the demographic data, comorbidities, and presence of urosepsis at admission.

There were no statistically significant differences between the two groups in age, sex, environment of origin, and comorbidities. All patients in both groups had at least one comorbidity. A statistically significant difference was found in favor of the CoV group related to the presence of urosepsis upon admission. Table 2 shows the presence of urinary catheters according to type.

In the CoV group, there were 17 (77.27%) patients with urinary catheters and lumbar drains, including 5 with urethral catheters, 4 with JJ stents, 6 with percutaneous nephrostomies, 1 case with ureterostomy splints, and 1 case with a lumbar drain tube inserted during a previous admission for drainage of a renal abscess. In the non-CoV group, 56 (84.85%) patients were carriers of urinary catheters, including 15 with urethral catheters, 30 with JJ stents, 9 with percutaneous nephrostomies, 3 with suprapubic permanent cystostomy, and 1 with a lumbar drain tube that was also inserted during a previous hospitalization for drainage of a renal abscess. There was a significant difference between the two groups regarding the presence of double-J stents in favor of the non-COVID group. All other parameters in the two groups, presented in Table 2, were similar. In Table 3, we comparatively present the risk factors studied for the two groups.

There were no significant differences between the two groups in terms of hospitalizations in the last 180 days, antibiotic therapy in the last 180 days, and the number and type of urologic interventions performed before the occurrence of the MDR UTI, with the only exception being for the previous transurethral resection of the prostate, which was significantly higher in the CoV group.

A significant difference was seen between the two groups in terms of days of hospitalization, with the CoV group having more days of hospitalization. In the CoV group there were 12 different types of urological maneuvers compared with the 14 in the non-CoV group. Interventions involving transurethral maneuvers (including ureteral catheterization) or urethral catheterization were the most common procedures preceding the MDR ITU in both groups. With open nephrectomy being the one exception for each group, all urological procedures were endoscopic or percutaneous. The two nephrectomies were performed for hemostatic purposes; in the CoV group, it was for a pyelocaliceal urothelial tumor, and in the non-CoV group, it was for spontaneous renal rupture with perirenal hematoma and hemorrhagic shock. The insertion of JJ stents was performed in both groups for infected ureterohydronephrosis. Percutaneous nephrostomies were performed for acute or chronic obstructive renal failure.

In the CoV group, six different types of MDR bacteria from the gut microbiota were found in the urine of the patients, compared to seven types in the non-CoV group. The types and numbers of gut microbiota in each group are shown in Table 4 and Figure 1.

The majority of MDR UTI bacteria identified in the urology clinic were, in order of incidence, *E. coli*, *Pseudomonas aeruginosa*, and *Klebsiella* spp. for the CoV group, and *E. coli. Klebsiella* spp., and *Pseudomonas aeruginosa* for the non-CoV group. They accounted for over 85% of all MDR UTI bacteria identified during the study period. Gram-negative bacilli accounted for 95.55% of all bacteria in the CoV group and 92.4% in the non-CoV group, while Gram-positive cocci (enterococci) accounted for only 4.5% and 7.6% in the CoV and non-CoV group, respectively. The sensitivity of the Gram-negative bacilli to the tested antibiotics is shown in Table 5 and Figure 2.

Enterococci were sensitive to vancomycin and amikacin in all cases. In Gram-negative bacilli, only piperacillin/tazobactam, imipenem, and meropenem were active at percentages greater than 75%. Most of these microorganisms showed resistance to cephalosporins (61.9–81% for the CoV group and 63.9–83.6% for the non-CoV group) and quinolones (71.4–76.2% for the CoV group and 80.3–82% for the non-CoV group).

## 3. Materials and Methods

We performed a case–control study on COVID-19-infected and non-CoV adult patients with MDR UTIs associated with gut microbiota that were admitted to the urology clinic of the “Dr. C.I. Parhon” Teaching Hospital in Iasi, Romania, between March 2020 and August 2022; these two groups of patients constituted the study group. For the control group, we consecutively added non-CoV patients with MDR UTIs at a ratio of 3 control patients to 1 study case admitted to the urology clinic in the same period. We considered MDR urinary infections to be where the bacteria were resistant to at least one antibiotic from three different classes [7,8].

In this study, 88 patients with confirmed MDR UTIs were chosen based on the following inclusion criteria: (i) suggestive clinical syndrome (dysuria or pollakiuria, with an exception for patients with urinary catheterization in whom clinical syndrome is defined by the presence of fever, chills, or an altered general condition); (ii) pyuria (≥10 leukocytes/mm^3^ in homogenized urine); and (iii) isolation of the gut bacteria in urine cultures (≥10^5^ CFU/mL). We excluded patients with enteric microorganism colonization (CFU > 10^5^ without symptoms) and those with a CFU < 10^5^/mL.

Urine cultures were obtained from all patients admitted to the clinic in the first hours after admission before empiric antibiotic therapy was started. Each urine culture was performed and interpreted by one of the two microbiologists in our clinic. The seeded volume was 10 μL of undiluted urine sampled with a calibrated disposable loop by the spreading method, which was then Gram stained and incubated aerobically at 37 °C overnight for 24 h on a gelatin culture medium and MacConkey medium, and finally followed by a qualitative assessment of bacteriuria. On the second day, an antibiogram was performed using the MicroScan Walkaway DxM1040, an automated analyzer from Beckman Coulter (Indianapolis, U.S.) with different antibiotics, depending on whether the bacteria were Gram-negative bacteria or Gram-positive cocci. The antibiogram method uses the MIC (minimal inhibitory concentration) breakpoint from the CLSI American Guidelines. The reference gut microbiota used by our laboratory were *Escherichia coli* ATCC 25922, *Pseudomonas aeruginosa* 27853, and *Staphylococcus aureus* ATCC 25923. If a UTI was suspected during hospitalization, new urine cultures were performed to detect MDR bacteria. CoV patients were diagnosed with a PCR test performed at the time of hospitalization or were known to have CoV before admission to our clinic. Until the test results were available, patients were placed in an isolation compartment for suspected CoV patients. If the test was negative, patients were transferred to the clinic. If the test was positive, the patients were transferred to the hospital’s dedicated CoV unit, where special isolation, hygiene, and protective conditions in addition to specific treatments were in place.

All study data were obtained from the electronic registry of the urology clinic and the analysis of the microbiology laboratory. All patients diagnosed with CoV during the study period with gut microbiota MDR UTIs were included in the study. These patients had no previously diagnosed CoV infection or long CoV. None of these patients had been vaccinated against CoV. Patients with MDR UTIs associated with other types of gut microbiota (one case with *Staphylococcus aureus* and one with *Candida* spp.) were excluded from the study. The same criteria were used to form the control group until a patient ratio of 3 to 1 in favor of the control group was reached. This ratio was chosen due to the expected small study group in order to increase the statistical power of the comparative tests.

After forming the two groups, we first comparatively analyzed age, gender, environment of origin, and the presence of comorbidities (diabetes mellitus (DM), neoplasms, chronic kidney disease, heart failure, anemia, hypertension (HTN), and urosepsis secondary to the MDR UTI). We then compared the risk factors of MDR UTIs: the presence of urinary catheters (urethral Foley catheters, JJ stents, nephrostomy tubes, cystostomy catheters, and lumbar drains), hospitalization in the last 180 days, and antibiotic prophylaxis in the last 180 days. We considered 180 days rather than 90 days as in other studies because there were patients with an MDR UTI that had a JJ stent inserted for more than 90 days and up to 180 days.

We analyzed whether there were urological maneuvers in the last 180 days and the type of procedures that were performed. We comparatively examined the types of urinary tract bacteria from the two groups and their sensitivity spectrum. Because there were two types of bacteria, Gram-negative bacilli and Gram-positive cocci, we analyzed their incidence and the sensitivity spectrum of the two groups separately.

Statistical analysis: Quantitative variables were described as mean and standard deviation, and their comparison was performed using Student’s *t*-test, with the threshold of *p* less than 0.05 being statistically significant. Qualitative variables were expressed as percentages, and their comparison was performed using the chi-square test and Fisher’s exact test, with *p* < 0.05 being statistically significant.

## 4. Discussion

The results of our study show an increased incidence of MDR UTIs associated with gut microbiota in CoV patients (23.16%) that were admitted to the urology clinic during the COVID-19 pandemic. Risk factors did not differ from those of non-CoV patients in the control group, but CoV patients with MDR UTIs were at increased risk for developing urosepsis and for a longer length of hospital stay. Patients in both groups had a wide range of urological interventions in their medical past, especially transurethral maneuvers (including ureteral catheterization). There were not significant differences between groups, as most patients were carriers of long-term urinary catheters, of which many of them had JJ stents, especially in the non-CoV group. The higher incidence of JJ stent carriers in the non-CoV group can be explained by the fact that these patients were admitted without restriction during the pandemic, and thus they could be diagnosed by a urine culture during hospitalization. The MDR germs involved were the same in both groups, with the most common being *E. coli*, *Klebsiella*, and *Pseudomonas*, and had the same spectrum of sensitivity with increased resistance to cephalosporins and quinolones.

The main limitation of this study is the relatively small group of CoV patients with MDR UTIs associated with gut microbiota that were included in the study, which could reduce the power of the comparative tests and result in a small difference between the studied characteristics of the two groups. Another limitation of this study is that some patients who underwent surgery in the urology clinic had, after discharge, urine cultures performed in other laboratories, and it is possible that some might have been diagnosed with an MDR UTI but were not referred to our clinic. Another explanation would be that some operated patients who had urinary tract colonization with MDR intestinal bacteria remained oligosymptomatic and did not return for treatment, and thus they remained undiagnosed. Therefore, it is probably more difficult to determine the real incidence of MDR UTIs with intestinal pathogens in all CoV patients. CoV patients with MDR UTIs who were admitted in our clinic were those who required urological procedures; therefore, the calculated incidence applies to these types of patients.

The incidence of MDR UTIs in CoV patients in our study was higher than the rate reported by other studies of MDR UTIs in hospitalized patients during the pandemic [2,4,8,9], but was similar to the rate reported by other studies of MDR UTIs in CoV patients [3,6]. This increased incidence in our clinic can be explained by the dramatic decrease in the scheduled elective admissions, being up to 50% monthly. Admissions were almost exclusively limited to urological emergencies [10,11], including urosepsis, patients with multiple comorbidities [3,12], neoplasms, permanent bladder catheter carriers, and other risk factors for the occurrence of an MDR UTI. Another explanation for the increased incidence in the CoV group is probably the additional risk of the occurrence of MDR urinary tract infections in CoV patients with urinary colonization (significant bacteriuria) with MDR enteric pathogens due to the impairment of the immune response by CoV [13], as evidenced by the higher incidence of urosepsis cases. It should be noted, however, that only two MDR UTIs were developed during the hospitalization of the CoV patients, demonstrating that the strict measures of isolation and protection from CoV were beneficial and helped prevent superinfection due to MDR bacteria [7,14]. The other patients were already colonized or had MDR UTIs, and the infection with CoV was contracted later. When the incidence of MDR UTI occurrence was calculated in the 77 patients with CoV but without MDR UTIs on admission, it was only 2.6% (2 patients out of 77), which is similar to the low incidence that has been reported in other studies [15,16]. The incidence was low even though some CoV patients were temporarily admitted to the intensive care unit [17,18].

The demographic characteristics were similar in both groups regardless of sex, age, and environment of origin, and the age of all these patients was over 60 years. With an older age and the necessity of urological maneuvers, the risk of developing an MDR UTI is increased. The fact that patients in both groups had at least one of the studied comorbidities may demonstrate the potential for compromising immune functions and increased susceptibility to colonization of the urinary system with MDR bacteria.

The risk factors analyzed were present in equal percentages, with our study highlighting the increased incidence related to permanent urinary catheters [19], urological maneuvers, and curative or prophylactic antibiotic therapy in the previous 6 months [20,21]. All operated patients also had antibiotic therapy, making it difficult to determine the separate involvement of urological maneuvers or antibiotic therapy in the occurrence of MDR UTIs; both probably play a role. The presence of diabetes, renal failure, anemia, and neoplasia has also been noted in other studies [22,23]. The longer hospital stay in CoV patients can be explained by the additional risk of CoV infection, which prolonged the evolution of the MDR UTI [16,24], and by the fact that more CoV patients presented with urosepsis, and therefore had more severe forms of MDR UTIs. Therefore, the evolution of CoV patients was longer due to the aggravation of the MDR UTI due to superinfection with CoV.

We found a large variety of previous urological interventions in MDR UTI patients either with or without CoV. The difference that we found in favor of CoV patients, which is related to transurethral resections of the prostate, can be explained by the very small number of patients who were admitted during the pandemic for this type of scheduled operation in our clinic. In all other types of operations, no difference was found; this is also true when considering the total number of operations performed before the appearance of MDR infections. The small number of previous open surgery operations is explained by the fact that the vast majority of scheduled open operations were postponed in our clinic during the pandemic.

*E. coli* was the most common MDR bacteria found in both groups, which is similar to findings in other studies [5,25], and was followed by *Klebsiella* spp. and *Pseudomonas aeruginosa*; these three bacteria accounted for over 80% of MDR UTIs. *E. coli* occurred in less than 50% of cases, compared with a 70–80% incidence in community infections [26]. In our study, Klebsiella and Pseudomonas occurred more frequently, whereas *Enterococcus* occurred less frequently compared with other studies [27,28]. Based on the bacteria type, the frequency of occurrence was almost similar, which was expected because most bacteria were present at the time of CoV infection, and thus had the same risk factors for their development.

Antibiograms of the gut microbiota showed high resistance to cephalosporins and quinolones in both groups, as was found in other studies [29], which is probably because of their frequent use in the treatment of urinary tract infections both in ambulatory and in hospitalized patients. Other authors believe that the abusive use of antibiotics in both humans and animals plays an important role in the occurrence of MDR infections both in the community and in hospitals [29]. In our study, resistance to piperacillin–tazobactam, imipenem, and meropenem was found to exceed 10%, which is higher than in some other studies [28,29], but is in accordance with other recent studies conducted locally [30,31]. These results are important in light of the recommendations of the European Association of Urology, whose guidelines call for the adaptation of antibiotic therapy to the local sensitivity spectrum of bacteria [26].

In our study, we also pointed out the increased resistance of these bacteria to ampicillin, augmentin, trimethoprim–sulfamethoxazole, and nitrofurantoin, which are commonly prescribed by primary care physicians, infectious disease specialists, and outpatient urologists. Therefore, these antibiotics should be avoided in patients with UTIs and who have the previously described risk factors for an MDR UTI. Infectious disease physicians treating CoV patients with the previously described risk factors for the occurrence of an MDR UTI should actively screen for the presence of these microorganisms via a urine culture.

## 5. Conclusions

MDR UTIs in both CoV and non-CoV patients are a major problem because of increased morbidity and the associated costs. In our study, urological patients at high risk for MDR UTI were profiled. Both CoV and non-CoV patients who had urologic interventions, especially transurethral ones, who remained carriers of urinary catheters, such as, most commonly, JJ ureteral stents and percutaneous nephrostomies, and who had associated comorbidities were at increased risk for an MDR UTI. The most commonly encountered bacteria were *E. coli, Klebsiella* spp., and *Pseudomonas aeruginosa*, which showed increased resistance to cephalosporins and quinolones. CoV patients with MDR UTIs may have higher rates of urosepsis and hospitalization, but further multicenter studies with a larger number of patients in the study group are needed to validate our findings.

## Figures and Tables

**Figure 1 antibiotics-12-00973-f001:**
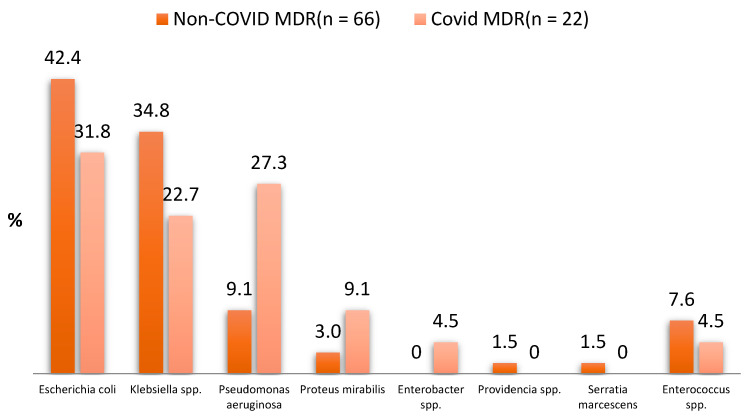
Types and numbers of enteral MDR bacteria in the two groups.

**Figure 2 antibiotics-12-00973-f002:**
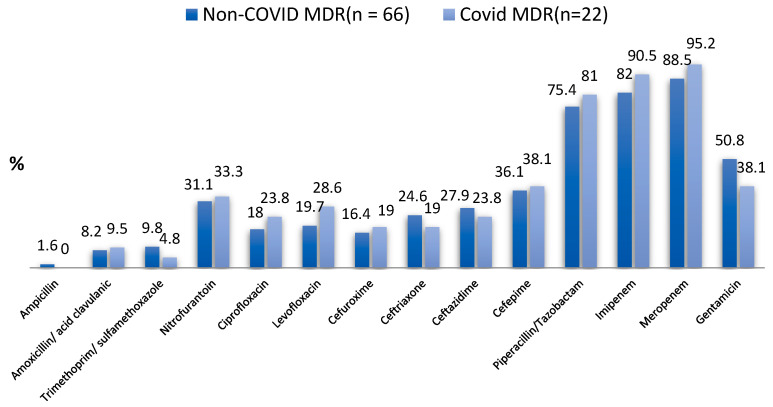
Antibiotic susceptibility of Gram-negative bacilli.

**Table 1 antibiotics-12-00973-t001:** Characteristics of the patients in the two groups.

		Non-COVID MDR UTI (*n* = 66)	COVID MDR UTI (*n* = 22)	*p*
Male		35 (53%)	12 (54.5%)	0.902 ^F^
Female		31 (47%)	10 (45.5%)	
Age (mean ± SD)	Male	63.31 ± 13.29	67.92 ± 6.99	0.136 ^t^
	Female	58.13 ± 18.69	65.20 ± 11.29	0.161 ^t^
Residence area (urban)		35 (53%)	10 (45.5%)	0.538 ^F^
Comorbidities				
1. Type 2 diabetes (DM)		20 (30.3%)	6 (27.3%)	0.787 ^F^
2. Neoplasia		27 (40.9%)	8 (36.4%)	0.706 ^F^
3. Kidney failure		26 (39.4%)	12 (54.5%)	0.214 ^F^
4. Heart failure		17 (25.8%)	3 (13.6%)	0.240 ^F^
5. Anemia		21 (31.8%)	7 (31.8%)	1 ^F^
6. Stroke sequelae		10 (15.2%)	3 (13.6%)	0.990^F^
7. Hypertension		28 (42.4%)	9 (40.9%)	0.901 ^F^
Urosepsis at the moment of admission		14 (21.2%)	12 (54.5%)	0.003 ^F^

MDR—multidrug-resistant; SD—standard deviation; t—Student’s *t*-test; F—Fisher’s exact test.

**Table 2 antibiotics-12-00973-t002:** Patients with urinary catheters at the time of diagnosis of MDR infection.

Urinary Catheters at the Time of Diagnosis	Non-COVID MDR UTI (*n* = 66)	COVID MDR UTI (*n* = 22)	*p*-Value for Chi-Square Test and Fisher’s Exact Test
Permanent urethral catheter	15 (22.7%)	5 (22.73%)	1
Permanent double-J ureteral catheter	30 (45.5%)	4 (18.2%)	0.023
Permanent nephrostomy catheter	9 (13.6%)	6 (27.3%)	0.190
Cutaneous ureterostomy catheter	2 (3.0%)	1 (4.5%)	0.999
Lumbar drain tube	1 (1.5%)	1 (4.5%)	0.440
Permanent cystostomy catheter	3 (4.5%)	0 (0.0%)	0.507
Total number of permanent urinary catheters	60 catheters in 56 patients (4 patients had 2 catheter types at the same time)	17 catheters in 17 patients	
Total number of patients with urinary catheters	56 (84.85%)	17 (77.27%)	0.413

**Table 3 antibiotics-12-00973-t003:** Risk factors for the occurrence of MDR infections.

	Non-COVID MDR UTI (*n* = 66)	COVID MDR UTI (*n* = 22)	*p*-Value for Chi-Square Test and Fisher’s Exact Test
Hospitalization in the last 180 days	51 (77.3%)	16 (72.7%)	0.665
Antibiotic therapy in the last 180 days	58 (87.9%)	17 (77.3%)	0.297
Hospitalization days (mean ± standard deviation)	6.09 ± 4.87	9.27 ± 4.92	0.010 ^t^
Types of urological interventions performed before the diagnosis of MDR infections			
1 TURP (transurethral resection of the prostate)	1 (1.5%)	3 (13.6%)	0.047
2 TURBT (transurethral resection of bladder tumors)	2 (3.0%)	0 (0.0%)	0.999
3 Percutaneous nephrostomy tube insertion	6 (9.1%)	5 (22.73%)	0.093
4 Nephrectomy	1 (1.5%)	1 (4.5%)	0.440
5 Urethral catheter replacement	8 (12.1%)	1 (4.5%)	0.440
6 Urethrotomy	3 (4.5%)	1 (4.54%)	0.440
7 Percutaneous nephrolithotomy (PCNL)	3 (4.5%)	0 (0.0%)	0.570
8 Percutaneous lumbar drainage	1(1.5%)	1 (4.5%)	0.440
9 Double-J catheter insertion	18 (27.3%)	2 (9.1%)	0.078
10 Double-J catheter replacement	14 (21.2%)	2 (9.1%)	0.338
11 Urethral catheter insertion	3 (4.5%)	2(9,1%)	0.425
12 Urethral dilatation	1 (1.5%)	0 (0.0%)	0.999
13 Percutaneous nephrostomy tube replacement	2 (3.0%)	2 (9.1%)	0.259
14 Ureterostomy double-J catheter replacement	1 (1.5%)	1 (4.5%)	0.440
Total urological maneuvers before the occurrence of MDR	64 (96.97%)	21 (95.45%)	0.734

MDR—multidrug-resistance; t—Student’s *t*-test.

**Table 4 antibiotics-12-00973-t004:** Types and numbers of enteral MDR bacteria in the two groups.

Types of MDR Bacteria	Non-COVID MDR UTI (*n* = 66)	COVID MDR UTI (*n* = 22)	*p*-Value for Chi-Square Test and Fisher’s Exact Test
*Escherichia coli*	28 (42.4%)	7 (31.8%)	0.379
*Klebsiella* spp.	23 (34.8%)	5 (22.7%)	0.290
*Pseudomonas aeruginosa*	6 (9.1%)	6 (27.3%)	0.066
*Proteus mirabilis*	2 (3.0%)	2 (9.1%)	0.259
*Enterobacter* spp.	0 (0.0%)	1 (4.5%)	0.250
*Providencia* spp.	1 (1.5%)	0 (0.0%)	0.999
*Serratia marcescens*	1 (1.5%)	0 (0.0%)	0.999
*Enterococcus* spp.	5 (7.6%)	1 (4.5%)	0.998
Total MDR infections	66	22	

**Table 5 antibiotics-12-00973-t005:** Antibiotic susceptibility of Gram-negative bacilli.

Type of Tested Antibiotic	Non-COVID MDR UTI (*n* = 61)	COVID MDR UTI (*n* = 21)	*p*-Value for Chi-Square Test and Fisher’s Exact Test
Ampicillin	1 (1.6%)	0 (0.0%)	0.999
Amoxicillin/Acid clavulanic	5 (8.2%)	2 (9.5%)	0.999
Trimethoprim/Sulfamethoxazole	6 (9.8%)	1 (4.8%)	0.671
Nitrofurantoin	19 (31.1%)	7 (33.3%)	0.853
Ciprofloxacin	11 (18.0%)	5 (23.8%)	0.541
Levofloxacin	12 (19.7%)	6 (28.6%)	0.542
Cefuroxime	10 (16.4%)	4 (19.0%)	0.747
Ceftriaxone	15 (24.6%)	4 (19.0%)	0.768
Ceftazidime	17 (27.9%)	5 (23.8%)	0.717
Cefepime	22 (36.1%)	8 (38.1%)	0.868
Piperacillin/Tazobactam	46 (75.4%)	17 (81.0%)	0.768
Imipenem	50 (82.0%)	19 (90.5%)	0.498
Meropenem	54 (88.5%)	20 (95.2%)	0.673
Gentamicin	31 (50.8%)	8 (38.1%)	0.314

## Data Availability

Not available.

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
