# Peer review of "Multidrug-Resistant (MDR) Urinary Tract Infections Associated with Gut Microbiota in CoV and Non-CoV Patients in a Urological Clinic during the Pandemic: A Single Center Experience"

_antibiotics, 2023, doi:10.3390/antibiotics12060973_

Round 1

Reviewer 1 Report

The study compares the profiles, and risk factors of Covid-19-infected and non-Covid-19-infected patients with multidrug-resistant urinary tract infections in the gut microbiota in a single center in Romania.

They conclude that there is no significant difference in the background profiles of both groups, and that sepsis may be slightly more common in the Covid-19 (CoV)-infected patients.

As methodology, the authors compared 22 Covid-19-infected patients with multidrug-resistant urinary tract infections to three times as many non-infected patients. We believe that the analysis population is too small to draw clinically meaningful conclusions.

In statistics, the repeated multiple testing procedure also makes it difficult to interpret the clinical significance of the items with p-values less than 0.05, since possibility of mere coincidence cannot be excluded.

Author Response

Thank you for reviewing the article. We agree that the study group is small for complex statistical analysis. However, the number of the patients in the study group was limited by the period of covid pandemic. We think that the data are still clinically relevant and useful in the research of the impact of covid pandemic on patients with MDR urinary tract infection. However, we have mentioned in the discussions of the study about the limitation related to the small number of cases. We have also downsized the conclusions and we added the following paragraph:” CoV patients with MDR-UTIs may have higher rates of urosepsis and hospitalization, but further multicenter studies with a larger number of patients in the study group are needed to validate our findings”.

Kindest regards,

Pavel Onofrei, MD, PhD

Reviewer 2 Report

In this study by Radu et al., 2023; the authors investigated the antimicrobial resistance profile of intestinal bacteria isolated in the urine of patients with Covid and patients without Covid, both with urinary tract infection problems. While the work is relevant to the field, there are some points that need to be better addressed in the manuscript, and a textual revision also needs to be done. Please see below:

1) Lines 3, 27, and where else it appears in the text: The term "germ" is inappropriate; it is too broad and includes organisms other than fungi and bacteria. This term cannot be used as a synonym for "microorganisms,” "urinary tract bacteria,” or "gut microbiota.” Therefore, I strongly suggest modifying the title and the text where this word appears.

2) lines 27, 30, and where else it appears in the text: please replace here and where else it appears "enteric flora" with "gut microbiota.”

3) line 30: this sentence needs to be clarified; please rewrite.

4) line 30: please include the number of patients with covid analyzed.

5) lines 30, 32 and where else it appears in the text: the terms "urinary infections with enteric flora" seem disconnected; please rewrite.

6) line 78: "]" seems unnecessary.

7) lines 84-86: please explain the methodology for culture-dependent methods in detail. Were serial dilutions made? What was the diluent? Was the spread plate method used? In duplicate? What is the incubation temperature for each medium? How many colonies were isolated from each sample/dilution or plate? What criteria were used?

8) line 112: please detail the procedure adopted for the antimicrobial susceptibility testing. Which recommendation was used to define whether the isolates were sensitive or resistant? What was the concentration of the antimicrobials used?

9) row 96: please specify which gut bacteria were used as the sampling cut-off reference.

10) lines 102-104: was the possibility of patients having had previous covid considered? Or the cases of long covid? This information is important for adequately dividing the groups. Are patients in both groups vaccinated against covid? How many doses were given?

11) About Tables 2, 3, 4, and 5: please specify which statistical test the values refer to in the last column.

12) lines 183-184: were found... "in the urine"?!

13) For me, tables 4 and 5 could be transformed into graphics, enriching the article’s presentation.

14) lines 283-285; 291-292: this period needs to be clarified whether the information presented refers to the literature or the article's findings; please rewrite.

15) line 301: what was the most problematic resistance profile found? Resistance to any last-generation antimicrobials?

16) There are several English problems throughout the text; I suggest a complete revision of the entire manuscript.

Please see the comments made previously.

Author Response

Dear reviewer, first of all, first of all, thank you for reviewing our paper. We answered on your notes as following:

1) Lines 3, 27, and where else it appears in the text: The term "germ" is inappropriate; it is too broad and includes organisms other than fungi and bacteria. This term cannot be used as a synonym for "microorganisms,” "urinary tract bacteria,” or "gut microbiota.” Therefore, I strongly suggest modifying the title and the text where this word appears.

We corrected

2) lines 27, 30, and where else it appears in the text: please replace here and where else it appears "enteric flora" with "gut microbiota.”

We corrected

3) line 30: this sentence needs to be clarified; please rewrite.

We corrected

4) line 30: please include the number of patients with covid analyzed.

We included

5) lines 30, 32 and where else it appears in the text: the terms "urinary infections with enteric flora" seem disconnected; please rewrite.

We corrected

6) line 78: "]" seems unnecessary.

We corrected

7) lines 84-86: please explain the methodology for culture-dependent methods in detail. Were serial dilutions made? What was the diluent? Was the spread plate method used? In duplicate? What is the incubation temperature for each medium? How many colonies were isolated from each sample/dilution or plate? What criteria were used?

We have corrected and added to the paragraph:

The seeded volume is 10 μL of undiluted urine sampled with a calibrated disposable loop by the spreading method, GRAM stained, and then incubated aerobically at 37℃ overnight/24 hours on gelatin culture medium and McConkey followed by the qualitative assessment of bacteriuria. Antibiogram on the second day was performed using MicroScan Walkaway DxM1040, an automated analyzer from Beckman Coulter. The antibiogram method uses the MIC (minimal inhibitory concentration) – breakpoint, CLSI – American Guidelines. Reference gut microbiota utilized by our lab were Escherichia coli ATCC 25922 , Pseudomonas aeruginosa 27853 and Staphylococcus aureus ATCC 25923.

Our microbiologists do not utilize diluted urine and considers CFU > 105 if the segment of the plate is almost entirely covered by bacterial colonies spread and confluent – so, a qualitative method, not quantitative method – standard procedure.

8) line 112: please detail the procedure adopted for the antimicrobial susceptibility testing. Which recommendation was used to define whether the isolates were sensitive or resistant? What was the concentration of the antimicrobials used?

The antibiogram method uses the MIC (minimal inhibitory concentration) – breakpoint, CLSI – American Guidelines.

The concentration of antimicrobial used is standardized e.g.: Imipenem - 10µg, Cefepime - 30µg, Levofloxacin 5µg etc.

9) row 96: please specify which gut bacteria were used as the sampling cut-off reference.

We corrected/added - The reference gut microbiota used by our laboratory were Escherichia coli ATCC 25922 and Pseudomonas aeruginosa 27853, Staphylococcus aureus ATCC 25923.

10) lines 102-104: was the possibility of patients having had previous covid considered? Or the cases of long covid? This information is important for adequately dividing the groups. Are patients in both groups vaccinated against covid? How many doses were given?

We added in the text - These patients had no previously diagnosed CoV infection or long CoV. None of these patients had been vaccinated against CoV.

11) About Tables 2, 3, 4, and 5: please specify which statistical test the values refer to in the last column.

For the tables there is the header indicating the tests.

P value for Chi-Square Tests Fisher's Exact Test

12) lines 183-184: were found... "in the urine"?!

We corrected

13) For me, tables 4 and 5 could be transformed into graphics, enriching the article’s presentation.

We transformed as graphics, but we don`t have the p value in the charts. We should kept them both of them.

14) lines 283-285; 291-292: this period needs to be clarified whether the information presented refers to the literature or the article's findings; please rewrite.

We corrected

15) line 301: what was the most problematic resistance profile found? Resistance to any last-generation antimicrobials?

*We had no bacteria resistant to all tested antibiotics, the most problematic profile found was bacteria who were sensitive to only Meropenem or Imipenem.

**During the period when the study was conducted, we hadn`t tested any last generation antimicrobials. For 2 months we test urinary tract bacteria on Zaficefta - , Zerbaxa and Recarbrio in our clinic.

16) There are several English problems throughout the text; I suggest a complete revision of the entire manuscript.

We proceeded for MDPI`s English editing service

Reviewer 3 Report

In the presented manuscript, the authors compared the profiles of two groups of patients with multidrug-resistant urinary tract infection: SARS-CoV-2 infected and SARS-CoV-2 non-infected. The patient were admitted in the Urology clinic of the Teaching Hospital in Iasi (Romania).

The Authors found that  Escherichia coli, Klebsiella spp. and Pseudomonas aeruginosa were the bacteria causing the most MDR infections in both patient groups. But MDR-UTI patients who had CoV-related symptoms showed a higher rate of urosepsis. Although the study analysed a relatively small group of patients, the data provided are valuable for fully understanding the effects of the Covid-19 pandemic.

I have only minor comments:

Lines 78-79, could you please clarify this sentence. Do you mean excluding those patients with enteric microorganisms colonization whose CFU was <10^5/ml?

 Lines 196-197 – should be … “for only 4,5% and  7,6% in the  CoV and non-CoV group, respectively”. (?)

Table 5. The title of the first column -“Number of MDR antibiotics”- seems inappropriate (Antibiotics/number of antibiotic susceptible isolates?).

 Lines 252-254. Shouldn't " the incidence of CoV occurrence" be replaced by "the incidence of MDR-UTI occurrence"?

Author Response

First of all, thank you for reviewing our paper and for your suggestions. We answered as following:

Lines 78-79, could you please clarify this sentence. Do you mean excluding those patients with enteric microorganism colonization whose CFU was <10^5/ml?

By enteric microorganism colonization we mean CHU over 105 without symptoms, therefore they do not have UTI.

Lines 196-197 – should be … “for only 4,5% and  7,6% in the  CoV and non-CoV group, respectively”. (?)

We corrected.

Table 5. The title of the first column -“Number of MDR antibiotics”- seems inappropriate (Antibiotics/number of antibiotic susceptible isolates?).

We modified with “Type of tested antibiotic” instead of number of MDR antibiotics

Lines 252-254. Shouldn't " the incidence of CoV occurrence" be replaced by "the incidence of MDR-UTI occurrence"?

We corrected.

Kindest regards,

Pavel Onofrei, MD, PhD

Round 2

Reviewer 1 Report

The addition of the limitation and changes to the conclusions would have made the paper scientifically valid.Although it is basically a negative study, we believe that in an unprecedented pandemic environment, the verification of a hypothetical association between urinary tract infections and novel coronavirus infections, as in this study, is also scientifically significant.

Reviewer 2 Report

Please substitute the term "germ" each time it appears. Line 232

Okay, about the English revision process.